# Blood-Based Biomarkers Predictive of Metformin Target Engagement in Fragile X Syndrome

**DOI:** 10.3390/brainsci10060361

**Published:** 2020-06-10

**Authors:** Mittal Jasoliya, Heather Bowling, Ignacio Cortina Petrasic, Blythe Durbin-Johnson, Eric Klann, Aditi Bhattacharya, Randi Hagerman, Flora Tassone

**Affiliations:** 1Department of Biochemistry and Molecular Medicine, University of California, Davis, Sacramento, CA 95817, USA; mjjasoliya@ucdavis.edu; 2Center for Neural Science, New York University, New York, NY 10003, USA; hlb248@nyu.edu (H.B.); ek65@nyu.edu (E.K.); 3MIND Institute, University of California, Davis, Sacramento, CA 95817, USA; icortina@ucdavis.edu (I.C.P.); rjhagerman@ucdavis.edu (R.H.); 4Department of Public Health Sciences, University of California, Davis, Davis, CA 95616, USA; bpdurbin@ucdavis.edu; 5NYU Neuroscience Institute, New York University School of Medicine, New York, NY 10016, USA; 6Center for Brain Development and Repair, Institute of Stem Cell Science and Regenerative Medicine, Bangalore 560065, India; aditi@instem.res.in; 7Department of Pediatrics, University of California Davis, Sacramento, CA 95817, USA

**Keywords:** biomarker, MMP9, RAS, metformin, Fragile-X syndrome

## Abstract

Recent advances in neurobiology have provided several molecular entrees for targeted treatments for Fragile X syndrome (FXS). However, the efficacy of these treatments has been demonstrated mainly in animal models and has not been consistently predictive of targeted drugs’ response in the preponderance of human clinical trials. Because of the heterogeneity of FXS at various levels, including the molecular level, phenotypic manifestation, and drug response, it is critically important to identify biomarkers that can help in patient stratification and prediction of therapeutic efficacy. The primary objective of this study was to assess the ability of molecular biomarkers to predict phenotypic subgroups, symptom severity, and treatment response to metformin in clinically treated patients with FXS. We specifically tested a triplex protein array comprising of hexokinase 1 (HK1), RAS (all isoforms), and Matrix Metalloproteinase 9 (MMP9) that we previously demonstrated were dysregulated in the FXS mouse model and in blood samples from patient with FXS. Seventeen participants with FXS, 12 males and 5 females, treated clinically with metformin were included in this study. The disruption in expression abundance of these proteins was normalized and associated with significant self-reported improvement in clinical phenotypes (CGI-I in addition to BMI) in a subset of participants with FXS. Our preliminary findings suggest that these proteins are of strong molecular relevance to the FXS pathology that could make them useful molecular biomarkers for this syndrome.

## 1. Introduction

Fragile X Syndrome (FXS) is the leading single-gene cause of Autism Spectrum Disorder (ASD) and intellectual disability (ID) caused by a CGG trinucleotide expansion in the *FMR1* (Fragile X mental retardation 1) gene. Remaining FXS cases that do not meet the criteria of ASD diagnosis demonstrate many behaviors that go along with the broader autism spectrum [1,2,3,4,5,6,7]. The overlap between FXS and ASD diagnoses and symptoms underscores the importance of FXS as a model population for ASD therapeutic development. Patients with FXS are at increased risk for ID, social anxiety and withdrawal, social deficits with peers, abnormalities in communication, unusual responses to sensory stimuli, stereotypic behavior, gaze aversion, inattention, impulsivity, hyperactivity, aggression, and self-injurious behaviors [5]. In addition, FXS is also associated with an increase in Body Mass Index (BMI), and a subgroup of patients with FXS can develop severe hyperphagia, obesity, and hypogonadism or delayed puberty [8,9]. Currently, there are no FDA-approved pharmacological therapies for FXS, despite its status as an orphan disease and its potential importance to also inform about ASD treatments. In contrast, there is a veritable wealth of preclinical studies in FXS in animal models, including mice, rats, Drosophila, zebrafish, and C. elegans. Over the past two decades, these preclinical models have yielded many insights into the molecular mechanisms of FXS as well as tangible leads for pharmacological intervention. However, translation into human systems has not been successful to date. Several of the targeted treatments do report some positive results but when combined across full cohorts, they do not report a consistent therapeutic efficacy in primary outcome measures in clinical trials [5]. The presence of a subset of responders with comorbid conditions with FXS clearly argues for a heterogeneous patient population that needs to be identified and stratified, and then potential treatments need to be devised. Therefore, it is of critical importance to find biomarkers that could help in patient stratification and can be used as a companion diagnostic for prediction of therapeutic efficacy.

Previous studies in brains of *Fmr1 KO* mice, the FXS mouse model, and in humans with FXS have demonstrated an increase in Matrix metalloproteinase-9 (MMP9) [10,11,12,13]. MMP9 is known to play an important role in normal brain development; indeed, its dysregulation is characterized by aberrant brain development and is implicated in the pathophysiology of various neurodegenerative and neurodevelopmental disorders, including FXS, ASD, bipolar disorder, and schizophrenia [14,15,16]. MMP-9 is an endopeptidase that cleaves extracellular matrix, cell adhesion molecules, cell surface receptors, cytokines, growth factor, and other proteases for synaptic reorganizations [14]. Knock out of MMP9 in FXS mice ameliorated several FXS behavioral phenotype [12]. The use of the antibiotic minocycline reduces MMP9 levels, restores normal dendritic spine development, and rescues abnormal behavior in FXS mice model [17,18,19]. Further, minocycline treatment in humans reduces MMP9 levels in most of the participants and is associated with improvement in behavior [10,20,21,22]. Taken together, MMP9 has the potential for being used as a molecular biomarker to determine therapeutic efficacy.

Although FXS is known to have imbalances in protein synthesis as a core molecular phenotype, this knowledge has not been leveraged by many studies aimed to develop molecular biomarkers [23,24]. Recently, our de novo brain-based proteomic screen study identified several proteins, including Ras (RAS), Hexokinase 1 (HK1), and Aconitase 2 (ACO2), with decreased expression levels in the *Fmr1 KO* hippocampus and in the peripheral blood of individuals with FXS [25]. We also observed a significant increase in Ras expression in FXS mice treated with metformin. Metformin treatment in *Fmr1 KO* mice was shown previously to rescue multiple FXS phenotypes [26]. Metformin is a type 2 diabetes drug that can also improve obesity and excessive appetite and has recently emerged as a candidate drug for targeted treatment of FXS based on animal studies showing rescue of multiple phenotypes in the FXS animal models [26,27]. Specifically, metformin treatment in the *Fmr1 KO* mice rescued aberrant social behavior, grooming, dendritic spine morphology, and hippocampal synaptic plasticity. In the fly *Drosophila melanogaster* FXS model, the use of metformin rescued and restored memory deficits [26,27].

In a recent study, we have utilized metformin in the clinical treatment of several individuals with FXS between the ages of 4 and 58 years and have found benefits not only in lowering weight gain and normalizing appetite but also in language and behavior [28].

In this study, we have investigated whether plasma abundance levels of RAS, HK1, and MMP9 were sensitive to metformin treatment in individuals with FXS and whether they correlated with changes in commonly used clinical measures particularly Clinical Global Impression (CGI) and Body Mass Index (BMI).

## 2. Method

### 2.1. Participants

Biological samples from 17 participants, including five females and 12 males with an age range of 2–8 and 1–67 years, respectively, were included in the study. Participants’ specimens were collected under protocols approved by the UC Davis Institutional Review Board upon consenting (IRB protocol: 1068417). Detailed information of each participant, including age, gender, mutation category (hypermethylated full mutation or size/methylation full mutation mosaic), metformin dose, duration of treatment, and other treatments, is listed in Table 1.

### 2.2. FMR1 Molecular Measures

Genomic DNA was isolated from 3 mL of peripheral blood following standard procedure (Qiagen, Germantown, MD, USA). Fragile X DNA testing, including CGG allele sizing and methylation status, was performed using both PCR and Southern blot analysis as previously reported [29,30]. Methylation status and Activation ratio in females were determined on the Southern blot as described in [31]. *FMR1* mRNA levels were measured by real time qRT-PCR as previously described [32].

### 2.3. MMP9 Expression Levels

Plasma samples were isolated from blood collected in EDTA tubes. Briefly, EDTA-containing collecting tubes were centrifuged at 2000 rpm for 10 min at 4 °C within 2 h of the blood draw. Plasma was collected without disturbing the interface, aliquoted in cryovials, and stored at −80 °C. Plasma samples were thawed to room temperature before running the assay.

Two analytes, MMP9 and MMP2, were measured in human plasma using ‘Human MMP Magnetic Bead Panel 2′ (HMMP2MAG-55K, Millipore, Sigma, St. Louis, MO, USA) as per the manufacturer’s instruction. Briefly, plasma samples were diluted 1:40 with the supplied assay buffer. Standards and samples were incubated with MMP9 and MMP2 beads for 2 h at room temperature. This was followed by incubation with detection antibody and streptavidin-phycoerythrin for 1 h and 30 min, respectively. Plates were read on Bio-Plex^®^ 200 Systems, Luminex xMAP technology (BioRad, Hercules, CA, USA). MMPs concentration was determined using a five-parameter logistic curve-fitting method on Bio-Plex Manager Software version 6.1. MMP9 concentration was normalized to MMP2 concentration to correct for plate-to-plate variability.

### 2.4. Western Blotting

Plasma samples were thawed, lysed, and denatured in Laemmli buffer containing protease inhibitors at a ratio of 3 μL sample to 100 μL buffer. 19 μL was loaded and analyzed by Western blotting—HK1 #2024 antibody 1:1000 dilution and RAS #3965 1:1000 dilution (Cell Signaling, Danvers, MA, USA). Each blot contained patient samples before and after treatment and was run on 4–12% BOLT NuPage gels (ThermoFisher, Waltham, MA, USA), transferred onto nitrocellulose using the iBLOT2 system (ThermoFisher, Waltham, MA, USA), and imaged using chemiluminescence on a FluorChem E (Protein Simple, San Jose, CA, USA). Blot images in TIFF format were shared with an offsite-analyst (who was blind to lane identities) on a secure online server. Images were quantified on Image J (NIH, Bethesda, MD, USA) by drawing a constant pixel size box over a band of interest and integrated density measured along with background intensity immediately below the band. Only lanes where the signal is at least twice the strength of background was quantified. All proteins were normalized to total protein level using MEMcode (Thermo Scientific, Waltham, MA, USA). MEMcode files were analyzed in the same session as yoked sets. Data were blindly analyzed and quantified using densitometry.

### 2.5. Clinical Measures

Clinical measures of study participants involved Clinical Global Impression Scale-Improvement (CGI-I). CGI-I is a seven-point scale that requires the clinician to assess how much the patient’s illness has improved or worsened relative to a baseline state at the beginning of the intervention. It is rated as follows: 1 = very much improved since the initiation of treatment; 2 = much improved; 3 = minimally improved; 4 = no change from baseline; 5 = minimally worse; 6= much worse; 7 = very much worse since the initiation of treatment [33].

Further, Body Mass Index (BMI) was measured before and after treatment. Patients were classified at the baseline visit as underweight, normal, overweight, and obese, depending on their age. For children at the age of 2–16 years, BMI less than 14 = underweight; BMI between 14 and18 = normal; BMI between 18 and 20 = overweight; BMI greater than 20 = obese (https://www.cdc.gov/healthyweight/images/assessing/growthchart_example1.gif). For adults older than 16 years, BMI less than 18 = underweight; BMI between 19 and 24 = normal; BMI between 25 and 29 = overweight; BMI greater than 29 = obese (https://www.nhlbi.nih.gov/health/educational/lose_wt/BMI/bmi_tbl.pdf). BMI was calculated again at the follow up visit and compared to that of the baseline visit. We defined a “normalized” BMI as an underweight patient gaining weight, a normal weight patient maintaining a normal weight, or an overweight patient losing weight. Therefore, any patient that either moved towards a healthier BMI or maintained a healthy BMI was considered “normalized” for subsequent BMI analyses.

### 2.6. Statistical Analysis

Ratios of protein levels to baseline were log2 transformed prior to analysis. The change in protein levels over time was estimated and tested using linear models that included age and visit interval as covariates. The association of overall BMI change with change in protein level was estimated and tested using linear models that included log2 change in protein level, age, visit interval, and baseline BMI as covariates. Analyses were conducted using R version 3.6.0 (26 April 2019).

## 3. Results

### 3.1. Participants

Metformin-treated participants with FXS were confirmed to have a full mutation with an *FMR1* allele with greater than 200 CGG repeats. Seventy-six percent (*n* = 13), including three females and ten males, had a hypermethylated full mutation, while 24% (*n* = 4), including two females and two males, were mosaics for the presence of both methylated and unmethylated alleles (Table 1). Among the mosaics, three, two males and one female, were methylation mosaics (methylated and unmethylated alleles spanning the entire expanded range), and one female was size mosaics (with a premutation (55–200 CGG repeats) and full mutation allele).

### 3.2. MMP9, HK1, and RAS Display Different Response to Metformin-Treatment

Our working hypothesis for this study was that there would be variability in the response to metformin among participants. We also anticipated that pharmacological interaction of metformin in FXS would be different as compared to its signature in diabetes or Polycystic ovarian syndrome. We therefore investigated if MMP9, HK1, and RAS could potentially track target and signaling pathway engagement of metformin in FXS.

We have previously reported that MMP9 protein levels are significantly higher in FXS patient plasma while RAS and HK1 protein level are significantly lower in FXS patients compared to age- and gender-matched typically developing (TD) individuals. [10,13,25]. Here, we measured the expression of these three targets, that are dysregulated in FXS, at both baseline and post metformin treatment.

Out of 17 participants that were treated with metformin, MMP9 levels were increased in 10 participants (eight males, two females) and decreased in six (one females, five males), while no to small changes were observed in one female participant, relative to each individual’s baseline. For HK1 expression, out of 17 participants, HK1 increased in seven (five males, two females), decreased in nine (seven males, two females), and no to small change was observed in one female. We were able to measure the RAS protein in 15 participants; RAS expression increased in 13 participants (eight males, five females, with one young female having no to a small increase), and we observed decrease in two of them, both males (Table 2; Appendix A).

We also looked at the aggregate response by determining change in log2 protein level from baseline to visit 1, adjusting for age and visit interval, using a linear model of change. We observed increase in MMP9 (*n* = 17) and RAS (*n* = 15) expression, while expression of HK1 (*n* = 17) decreased post treatment. However, none of these changes reached significance (Table 3).

### 3.3. Correlation of Molecular Measures with Clinical Measures

In this study, we also investigated the correlation of protein abundance with clinical measures recorded, namely BMI and CGI-I. Importantly, we found that 10 out of 11 participants showed a “BMI normalization” (defined as gaining weight if they started underweight, maintaining normal weight if they began at normal weight, or losing weight if they began metformin as overweight), while one participant began as obese and gained weight after metformin treatment. The advantage of using BMI normalization instead of change in BMI is that we could account for positive weight benefits in both directions as participants began the study on both ends of the weight spectrum.

We then asked if there was any association between HK1, RAS, and MMP9/MMP2 and normalization of the BMI, which would indicate that metformin is acting on a known drug target and as such could be considered a pharmacodynamic marker. To analyze the association with the markers and BMI normalization, we compared 11 participants for whom BMI data was available and analyzed the relationship between the change in expression levels of MMP9, HK1, and RAS and BMI normalization. We considered “increase” in expression if there was a value over 1.05 in the first visit/baseline ratio; anything less than 0.95 was considered as a decrease while a value of 0.95–1.05 indicated no change. RAS was measured in 15 participants, 11 of which had BMI measurements (Table 2). RAS expression increased in 13 participants and decreased in two; of those with an available BMI, RAS expression increased in nine of them, and all experienced BMI normalization, while RAS decreased in one, who did not experience BMI normalization and RAS did not change in one participant that experienced BMI normalization. RAS had a negative correlation with BMI change overall (linear model of change, adjusted for age and visit interval, 95% CI, *p* = 0.06) (Table 4). For MMP9 expression, there was no obvious relationship between BMI normalization and MMP9 expression, but there was a relationship between overall decreased BMI and increased MMP9 expression (linear model of change, adjusted for age and visit interval, 95% CI, *p* = 0.02). There was a trend level association between BMI and HK1 changes (linear model of change, adjusted for age and visit interval, 95% CI, *p* = 0.15) (Table 4). Out of 11 participants, seven showed an increased HK1expression and six moved to normal BMI, while four participants showed decreased HK1 expression and BMI normalization, and one participant showed a decrease in HK1 expression with no change in the BMI (Table 2).

We then did a similar correlation of our triplex protein panel with CGI-I to evaluate whether these molecular biomarkers could be predictive of an overall clinical improvement and thus, metformin efficacy on FXS. Out of the 17 participants treated with metformin, we had CGI-I for 16 participants, and none exhibited worse symptoms on the CGI-I scale. Only two had no positive change (CGI-I score of 4, both females); the other 14 had some positive change (CGI-I score 1–3) (Table 2). Plasma MMP9 levels were measured in all 17 participants, and those showing increased expression levels had a positive score of 1–3. Further, participants with a decreased HK1 expression had a score of 1–3, and over 75% of participants showing an increase in Ras expression also had a positive score of 1–3 (Table 2). We also observed sex differences in protein levels; specifically, males who showed a decreased in MMP9 levels after treatment had an improved score, whereas females that had decreased MMP9 had a no improvement in score (Table 2). HK1 did not have a uniform response in females and an age-dependent response in males.

Males over 18 years of age showed an increased level resulting in a positive response (CGI-I score of 1), and under 18 years of age, a decreased level resulted in a positive response (CGI-I score of 1); there were no males with no response (CGI-I score of 4) (Table 2). 

### 3.4. Plasma Proteins Levels Are Affected by Processing Time

During our protocol optimization, we compared samples from multiple clinics, and we noted a difference in the final protein quality and sample integrity, despite the use of the same collection tubes (EDTA containing collection tubes). To accurately determine if plasma processing time had a measurable effect on the molecular biomarker measurement and therefore levels, we collected samples and left them at room temperature for different increments of time before processing—30 min, 2 h, 4 h, and 24 h.

For MMP9 measurement, we observed a time-dependent increase in MMP9 expression, with a significant twofold increase at 24 h (*n* = 7, *p* = 0.04) (Figure 1). For RAS and HK1 quantification using Western blotting, some samples required no optimization, while others required a higher detergent content and decreased time of transfer. We determined that the main difference between the samples was the time between collection of blood from participants and when the samples were processed and frozen. Empirically, the longer the plasma was allowed to sit before processing and freezing, the more standardization of protein characterization method was needed. Changes in RAS and HK1 expression are shown in **(**Figure 1).

We also developed stringent criteria for Western blot and band analyses that were done by a double-blinded experimentalist off site. Strict quality control guidelines were also instituted that were informed by test–retest rounds (see details in the methods). As opposed to our previous report (Bowling 2019), we adopted a whole protein normalization to account for secreted proteins that make up the bulk of the protein content of whole blood.

## 4. Discussion

Although FXS is the world’s leading inherited monogenic cause of autism spectrum disorder (ASD) and intellectual disability, currently, there is no cure or approved medication for the treatment of the underlying causes. In the past 10 years, several targeted treatments for FXS have been performed in human subjects and although they have reported some positive results, they did not report a consistent therapeutic efficacy in primary outcome measures [5]. These studies have highlighted the need for unbiased biomarkers to better measure and predict treatment outcomes [34,35]. In this study, we tested a triplex of proteins as molecular biomarkers for their efficacy in predicting stratification, target engagement, and treatment outcome individually or as a combinatorial barcode in participants with FXS treated clinically with metformin. Looking at the potential signaling pathways, metformin appears to be a good candidate for targeting several of the intracellular functions in neurons disrupted in FXS and therefore possibly rescuing several types of symptoms in individuals with FXS. As an off-label drug, metformin has high safety ratings and is currently being tested in a controlled trial in a larger cohort of patients with FXS (https://clinicaltrials.gov/ct2/show/NCT03479476). There are currently no predictive biomarkers for selecting FXS patients who are likely to benefit from metformin that correct the altered protein expression observed in FXS and relevant for the pathology.

Our findings show that Ras can predict target engagement and BMI normalization for metformin, whereas HK1 and MMP9 had positive associations with CGI-I. Therefore, deploying all three biomarkers together could help track patients with FXS, in whom chemical cascades commensurate with metformin are being engaged and can inform if drug action suitably results in clinical improvements. Further, because of the high percentage of overlap between FXS and ASD, these biomarkers could also be useful in the larger ASD population.

A key problem that we have identified is the lack of stringent standard operating procedures (SOPs) for sample collection and handling, which has important implications in using molecular readouts in multisite trials. This is further compounded by problems of lack of consistency in data analyses. There is a paucity of data collection and analysis processes discussed in depth for most large trials that pertain blood work and imaging. These procedures must be made available to the community to enable further trials.

A number of caveats have to be considered for small open label studies and important considerations for the BMI data: (a) there are likely subgroups of responders, as the population varied in age, sex, and initial BMI status, and (b) the treatment duration and dose varied greatly between participants, which introduces more variability into the dataset. A controlled double-blind placebo trial would address many of these concerns and provide clearer data. Although CGI-I is an informative but complex tool with mixed populations such as those included in this open label pilot study, it is well established that there are sex-dependent differences in FXS for cognition and behavior, as well as age-dependent performance and plasticity [36,37,38]. Therefore, a controlled enrollment trial with multiple different measures of cognitive and behavioral functioning for comparison would provide greater points of reference to test the relationship between efficacy in FXS and our potential biomarkers.

In summary, we have identified, in a small pilot cohort, usable molecular, blood-based biomarkers to track target engagement and treatment efficacy of metformin in FXS. We hope this result can be scaled into larger trial cohorts to help develop usable biomarkers for FXS patient stratification and treatment outcome measures.

## Figures and Tables

**Figure 1 brainsci-10-00361-f001:**
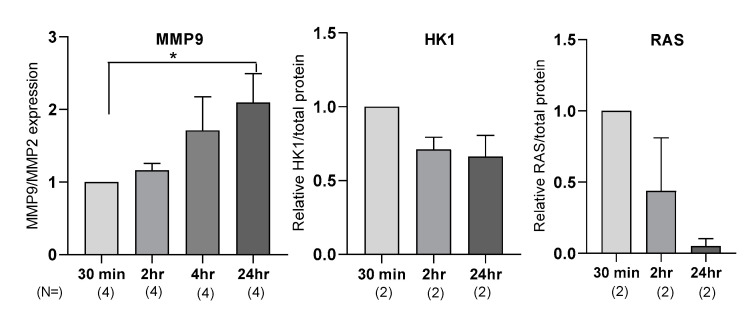
Sample processing time affects plasma MMP9, HK1, and RAS expression levels. Plasma MMP9, HK1, and RAS expression levels were measured at different time points: 30 min, 2 h, 4 h, and 24 h after blood draw. MMP9 expression was measured by luminex xMAP technology. Ras and HK1 expression were measured by Western blot. N represents number of independent biological replicates. Statistical significance was measured by Student’s t-test when *n* ≥ 3; * represents *p* < 0.05; error bars represents mean ± standard error of mean. Target protein expression levels were normalized to the 30 min time point to determine change in expression with time.

**Table 1 brainsci-10-00361-t001:** Demographic, molecular and medication information for participants included in the study. QD: once a day; BID: twice a day; TID: three times a day; QHS: every night at bedtime; QAM: every morning.

Participants	Age	Gender	Mutation Category	CGG Repeat	% Meth.	Activation Ratio	*FMR1* mRNA ± StErr	Metformin Dose	Duration of Treatment (Months)	Other Ongoing Treatments
Case 1	25	M	Meth mosaic	>200 (250–480)	78%		0.21 ± 0.004	1000 mg BID	28	Abilify 10 mg QD Effexor 75 mg QD
Case 2	10	M	Full mutation	>200			0	500 mg BID	15	Sertraline 50 mg QD Abilify 5 mg QHS Guanfancine 2 mg BID Minocycline 100 mg QD CBD and THC 2:1
Case 3	7	M	Full mutation	>200			0.13 ± 0.0001	500 mg BID	6	Clonidine 0.1 mg TID& 0.2 at QHS Adderall 2.5 mg TID Trileptal 420 mg BID
Case 4	1	M	Full mutation	>200			0.001 ± 0.00	150 mg BID	9	Melatonin 2 mg Sertraline 4 mg QD
Case 5	2	M	Full mutation	>200			0	50 mg BID	4	Melatonin 2 mg Sertraline 2.5 mg QAM
Case 6	23	M	Full mutation	>200			0	500 mg BID	16	Vitamin D Fish Oil Pravastatin 10 mg QD
Case 7	3	F	Size mosaic	28, >200 (54)	93%	0.38	0.95 ± 0.13	100 mg BID	7	Sertraline 0.2 mL (Zoloft 20 mg/mL)
Case 8	3	F	Full mutation	30, >200		0.5	0.83 ± 0.03	100 mg BID	1	Melatonin Multivitamin
Case 9	5	F	Full mutation	30, 240		0.52	1.01 ± 0.03	200 mg BID	9	Sertraline 20 mg QD Melatonin 5 mg QHS
Case 10	19	M	Full mutation	>200			0.05 ±0.00	500 mg QD	26	
Case 11	67	M	Full mutation	>200			0.07 ± 0.01	1500 mg TID	28	Sertraline 50 mg
Case 12	8	F	Meth. mosaic	31, >200 (360)	92%	0.42	0.26 ± 0.01	500 mg BID	1	
Case 13	2	M	Meth. mosaic	>200 (180–300)	82%		NA	500 mg BID	4	Sertraline 13 mL QD Multivitamins.
Case 14	10	M	Full mutation	>200			0	500 mg BID	7	Sertraline 25 mg Multivitamin Aripiprazole (Abilify) 2 mg Clonidine 01 mg qD + H15:H21
Case 15	8	M	Full mutation	>200			0	500 mg BID	1	
Case 16	2	F	Full mutation	29, >200		0.43	0.4 ± 0.01	50 mg BID	30	Melatonin 1 mg/Ml Sertraline HCL 2.5 mg QD Metformin 50 mg BID
Case 17	7	M	Full mutation	>200			0	500 mg BID	11	Guanfacine HCK Aripiprazole (Abilify) 5 mg QD Clonidine 0.1 mg QD

**Table 2 brainsci-10-00361-t002:** Patient molecular barcode representing change in MMP9, HK1, and Ras plasma expression levels; clinical measures including CGI-I post treatment; and change in BMI post metformin treatment relative to each patient’s own baseline. Blood samples for each participant before and after metformin treatment were processed within 2 h of collection time for plasma isolation. MMP9 expression was measured by Luminex xMAP technology. Ras and HK1 expression levels were measured by Western blot. Cut-off ±5% was used; increased expression was >5%, decreased expression was <5%.

Participants	Age	Gender	MMP9/MMP2 Changes	HK1 Changes	Ras Changes	CGI-I Score	BMI Move Towards Normal?
Case 1	25	M	0.46	1.08	3.92	2	yes
Case 2	10	M	1.09	0.65	-	2	NA
Case 3	7	M	0.81	0.73	1.1	1	stay normal
Case 4	1	M	0.65	1.08	1.37	3	stay normal
Case 5	2	M	1.64	0.65	-	2	yes
Case 6	23	M	0.25	6.75	1.22	1	yes
Case 7	3	F	1.75	0.99	1.1	2	yes
Case 8	3	F	0.96	1.01	2.07	4	NA
Case 9	5	F	0.82	1.14	2.6	4	NA
Case 10	19	M	1.41	1.34	1.07	3	yes
Case 11	67	M	5.36	0.94	0.53	2	no
Case 12	8	F	1.48	0.9	2.45	3	NA
Case 13	2	M	1.33	1.18	2.07	2	yes
Case 14	10	M	1.87	0.88	1.1	3	yes
Case 15	8	M	0.66	0.48	1.18	2	NA
Case 16	2	F	1.54	1.46	1.01	1	yes
Case 17	7	M	2.62	0.86	0.94	NA	NA

**Table 3 brainsci-10-00361-t003:** Linear model of change in protein level from baseline to post metformin treatment visit.

	Fold Change (95% CI)	*p*-Value
MMP9/MMP2	1.32 (0.569, 3.038)	0.494
HK1	0.793 (0.394, 1.595)	0.488
Ras	1.72 (0.918, 3.233)	0.084

**Table 4 brainsci-10-00361-t004:** Linear model of change in Body Mass Index (BMI) by change in protein level from baseline visit to post metformin visit.

	Regression Slope (95% CI)	*p*-Value
MMP9/MMP2	1.49 (0.235, 2.748)	0.0271
HK1	−1.33 (−3.298, 0.634)	0.148
Ras	−3.17 (−6.490, 0.143)	0.0572

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
