# Peer review of "Blood-Based Biomarkers Predictive of Metformin Target Engagement in Fragile X Syndrome"

_brainsci, 2020, doi:10.3390/brainsci10060361_

Round 1
Reviewer 1 Report
The clinical study by Jasoliya et al used previously-identified blood-based biomarkers (MMP9, RAS and HK1) in stratification of FXS patients and prediction of their responsiveness to metformin. Metformin has shown therapeutic potential in preclinical and small clinical studies and its further testing in patients with FXS is of paramount importance. This study explores an important idea of using blood biomarkers to select patients who will benefit from metformin. In addition to measuring blood biomarkers before and after metformin treatment, clinical phenotypes (CGI-I and BMI) were also assessed. The paper is well written, the data are clearly presented and extensively discussed. It is an important study that advances the understanding of FXS and therapeutic potential of metformin.
I do not have any major comments, it would be nice to see some examples of western blots.
Author Response
REVIEWER-1
Comment 1: it would be nice to see some examples of western blots.
Please see below examples of western blots and graphs of the ration post metformin/pre-metformin for HK1 and RAS complete with total protein stain (MEM Code) and N (number of samples).
Reviewer 2 Report
The manuscript entitled “Blood-based biomarkers predictive of metformin target engagement in Fragile X syndrome” is interesting. The authors tested hexokinase 1 (HK1), RAS (all isoforms) and Matrix Metalloproteinase 9 (MMP9), which were previously demonstrated as dysregulated in the FXS mouse model and in blood samples from patient with FXS. As mentioned in introduction, the authors have focused on “whether plasma abundance levels of RAS, HK1 and MMP9 were sensitive to metformin treatment in individuals with FXS and whether they correlated with changes in commonly used clinical measures particularly Clinical Global Impression (CGI) and Body Mass Index (BMI).” In Figure 1, the RAS expression levels were significantly decreased at 24 hours compared to 30 min. However, there is no p-value for this significant decrease – any reasons? I think “N” represents the number experiments in Figure 1. If my assumption on “N” is correct, then why the authors performed just experiments for 2 times in case of RAS and HK1 but four times in case of MMP9 – the authors need to explain the reasons for this issue. In Table 2 legend, the authors required to emphasize about the type of experiment they used to obtain the expression levels and also the details on how they derived the expression changes. In summary, the authors described potential blood-based biomarkers to track the treatment efficacy in metformin in FXS.
Author Response
REVIEWER-2
Comment 1: In Figure 1, the RAS expression levels were significantly decreased at 24 hours compared to 30 min. However, there is no p-value for this significant decrease – any reasons? I think “N” represents the number experiments in Figure 1. If my assumption on “N” is correct, then why the authors performed just experiments for 2 times in case of RAS and HK1 but four times in case of MMP9 – the authors need to explain the reasons for this issue.
The reason for the disparity in N number is sample availability. MMP9 analysis was performed on site at UC Davis and RAS and HK1 were performed at NYU, and therefore required several more steps of shipping and processing. It is unclear when we would be able to add to the N given that both institutions are currently shut down (COVID-19) and it is unclear when will be able to resume research. Usually, we do not include statistics for n=2, but in case it helps the reviewers, they are as follows using a student’s 2- tailed t-test.
Comment 2: In Table 2 legend, the authors required to emphasize about the type of experiment they used to obtain the expression levels and also the details on how they derived the expression changes
Expression levels were obtained as explained in the methods section of the manuscript – MMP9/MMP2 were measured by Luminex xMAP (Multi-Analyte Profiling) technology and HK1 and RAS were detected by Western blot as described in the methods. For the analysis samples were coded after collection in the clinic then processed blindly and analyzed. Fold change was generated by post metformin/pre metformin/ total protein. Samples were also randomly re-run and spot checked for consistency and remained consistent." We have made the suggested edits in the Table 2 legend (line# 212-217).
Reviewer 3 Report
Dear Authors,
I had the pleasure of reviewing the manuscript entitled “Blood-based biomarkers predictive of metformin target engagement in Fragile X syndrome” The manuscript is well-written and easy to follow. I have two minor comments, and they are as follows:
Comment 1 # Page 5, please state the full form of the term FMR1.
Comment 2 # Please put the detailed figure legend for the Figure. 1. Mention which experiment was performed, the number of study subjects, and the name of statistical method and P values.
After implementing these two minor comments, I would recommend this study to be published in the Journal, Brain Sciences.
Author Response
REVIEWER-3
Comment 1: Page 5, please state the full form of the term FMR1.
We have added the full form as suggested at page 3 (line# 53-54)
Comment 2: Please put the detailed figure legend for the Figure. 1. Mention which experiment was performed, the number of study subjects, and the name of statistical method and P values.
We have included the suggested explanations in the legend of Figure 1 (line # 302-309).